DATA RELEASE

# *Anopheles* sampling collections in the health districts of Korhogo (Côte d'Ivoire) and Diébougou (Burkina Faso) between 2016 and 2018

Paul Taconet[1], Barnabas Zogo[2], Dieudonné Diloma Soma[3,4], Ludovic P. Ahoua Alou[2], Karine Mouline[1], Roch Kounbobr Dabiré[2], Alphonsine Amanan Koffi[2], Cédric Pennetier[1,2] and Nicolas Moiroux[1,*]

1 MIVEGEC, Université de Montpellier, CNRS, IRD, Montpellier, France
2 Institut Pierre Richet (IPR), Institut National de Santé Publique (INSP), BP 1500, Bouaké, Côte d'Ivoire
3 Institut de Recherche en Sciences de la Santé (IRSS), BP 545, Bobo Dioulasso, Burkina Faso
4 Institut Supérieur des Sciences de la Santé, Université Nazi Boni, BP 1091, Bobo-Dioulasso, Burkina Faso

## ABSTRACT

Characterizing the entomological profile of malaria transmission at fine spatiotemporal scales is essential for developing and implementing effective vector control strategies. Here, we present a fine-grained dataset of *Anopheles* mosquitoes (Diptera: Culicidae) collected in 55 villages of the rural districts of Korhogo (Northern Côte d'Ivoire) and Diébougou (South-West Burkina Faso) between 2016 and 2018. In the framework of a randomized controlled trial, *Anopheles* mosquitoes were periodically collected by Human Landing Catches experts inside and outside households, and analyzed individually to identify the genus and, for a subsample, species, insecticide resistance genetic mutations, *Plasmodium falciparum* infection, and parity status. More than 3,000 collection sessions were carried out, achieving about 45,000 h of sampling efforts. Over 60,000 *Anopheles* were collected (mainly *A. gambiae* s.s., *A. coluzzii*, and *A. funestus*). The dataset is published as a Darwin Core archive in the Global Biodiversity Information Facility, comprising four files: events, occurrences, mosquito characterizations, and environmental data.

**Subjects** Ecology, Biodiversity, Taxonomy

**Submitted:** 03 May 2023

\* Corresponding author. E-mail: nicolas.moiroux@ird.fr

Preprint submitted at https://doi.org/10.31730/osf.io/j7ux8

Included in the series: *Vectors of human disease series* (https://doi.org/10.46471/GIGABYTE_SERIES_0002)

## DATA DESCRIPTION

### Background and context

Malaria is a major vector-borne disease, still affecting more than 200 million people and causing more than 500,000 deaths worldwide annually [1]. Malaria parasites are transmitted to humans by infected female mosquitoes of the genus *Anopheles*. Although malaria control efforts have led to a sustained decrease in the disease burden between 2000 and 2015 (mainly through the widespread use of long-lasting insecticidal nets (LLIN) [2]), progress is now stalling [1]. Vector resistance to insecticides, human population growth, and environmental changes are involved in such worrying trends. To reinvigorate progress, vector control (VC) strategies need to be tailored locally, building on a thorough knowledge of the local determinants of malaria transmission [3, 4]. To do so, it is essential to

characterize the entomological profile of malaria transmission at fine and operational spatiotemporal scales [3–5].

In this paper, we present a dataset derived from mosquitoes collected for a randomized controlled trial (RCT) as part of a project called REACT ('Insecticide resistance management in Burkina Faso and Côte d'Ivoire: research on vector control strategies'). Involving three scientific partners – the Institut de Recherche pour le Développement (IRD, France), the Institut de Recherche en Science de la Santé (IRSS, Burkina Faso (BF)), and the Institut Pierre Richet (IPR, Côte d'Ivoire (CI)) – the REACT project aimed to assess whether the addition of complementary VC strategies to LLINs reduces malaria transmission and provides additional protection against malaria in areas with a high level of insecticide-resistant vectors in rural BF and CI. In this frame, periodic entomological surveys aimed at characterizing the *Anopheles* populations were carried out between 2016 and 2018. Data on *Anopheles* species composition, abundance, biting behaviour, genetic mutations providing insecticide resistance, and infections caused by malaria parasites were collected. This paper aims to detail these data and the necessary contextual information to use them properly.

## Methods

### Study areas, temporal coverage, and sampling frequency

The study areas of the REACT project were the rural health districts of Korhogo (CI) and Diébougou (BF). Each study area covers about 2,500 square km. Both countries are endemic for *Plasmodium falciparum* malaria [1]. The main *Anopheles* species in these countries are *Anopheles arabiensis, Anopheles gambiae sensu stricto*, and *Anopheles coluzzii*, which belong to the *Anopheles gambiae* complex, and *A. funestus*, which is a member of the *Funestus* group [6–8]. In both countries, *Anopheles*' insecticide resistance has been reported for several decades [9, 10].

The Diébougou area is located in the South-West of BF, in the Sudanian bioclimatic region [11]. The climate is characterized by a dry season from October to April (including a 'hot' period from October to November and a 'cold' period from December to February) and a rainy season from May to September. The daily temperature ranges are 18–36 °C, 25–39 °C, and 23–33 °C in the cold dry, hot dry, and rainy seasons, respectively. The average annual rainfall is 1,200 mm. The natural vegetation is dominated by savannah trees interspersed with riparian forests. The main economic activity is agriculture (cereal cultivation), followed by artisanal gold mining, and charcoal and wood production [12, 13]. The leading VC tool in the Diébougou region is the LLIN, distributed universally by the government every 3–4 years since 2010 [14]. The last distribution before the REACT project was in July 2016.

The Korhogo area is located in the north of CI, also in a Sudanian bioclimatic region [11]. The seasonality of the climatology is relatively similar to that of Diébougou. Annual rainfall varies from 1,200 to 1,400 mm, while the average annual temperature varies from 21 °C to 35 °C. The natural vegetation is mainly a mixture of savannah and open forest. The region has a high density of hydraulic dams that allow for year-round agriculture. As for the Diébougou region, the main economic activity is agriculture (rice, maize, cotton). Similarly to Diébougou, the primary vector control tool is the LLIN, distributed universally by the government, as in BF, every 3–4 years since 2010 [15]. The last distribution before the beginning of the REACT project was in 2014. During the project, LLINs were distributed in the study villages in June 2017.



As mentioned previously, the REACT project consisted of an RCT. Within each area, several villages (27 in the BF area and 28 in the CI area) were selected at the beginning of the project according to the following criteria: accessibility during the rainy season, 200 to 500 inhabitants per village, and distance between villages greater than 2 km. In each village, several rounds of mosquito collections (surveys) were carried out using the Human Landing Catch (HLC) sampling method. In the BF area, seven entomological surveys were conducted over 15 months between January 2017 and April 2018. In the CI area, eight entomological surveys were conducted over 18 months between October 2016 and April 2018. As part of the RCT, VC tools complementary to the LLINs were deployed in selected, randomized villages in September 2017. Namely, the following VC strategies were deployed: (i) larviciding with *Bacillus thuringiensis israelensis* to target immature stages of *Anopheles* species, (ii) indoor residual spraying with pirimiphos-methy to target endophilic malaria vectors, (iii) information, education, and communication activities to improve LLIN use, and (iv) ivermectin administration to animals, a One Health approach to tackle zoophagic behaviour of malaria vectors and improve animal health. The associated data table in GigaDB [16] lists the villages included in this study: names, geographic coordinates, control and experimental villages. The dates of the entomological surveys were chosen to compare the entomological indicators before and after the deployment of VC strategies as part of the RCT, and to span over the typical climatic conditions of these tropical areas.

Figure 1 shows the geographic location of the villages, the dates of mosquito samplings, and key dates for VC interventions in the study areas.

### Sampling methods

The procedure for HLC consisted of the collection, using hemolysis tubes, of mosquitoes alighting on the exposed legs of people sitting on stools. Mosquitoes were collected from 17:00 to 09:00, both indoors and outdoors, at four sites per village. Collectors were organized into two teams of eight persons in each village; the first group collected from 17:00 to 01:00, and the second from 01:00 to 09:00. Collectors were rotated between indoor and outdoor collection sites every hour at each selected house (sites) to reduce potential collector bias. Indoor collection points were rooms that met the following criteria: being usually inhabited; quiet without excessive movement of people; open to the outside through a door or a window. The outdoor collection was conducted in areas usually occupied by people but sheltered from wind, traffic, and fires, and away from large meeting areas. The distance between collection sites was at least 50 m. The distance between indoor and outdoor collection points in one site was at least 10 m to minimize competition between mosquito collectors. Mosquitoes collected in individual hemolysis tubes plugged with cotton were stored in hourly bags.

### Morphological identification

All captured mosquitoes were morphologically identified in the field to their genus and, when possible, species according to established taxonomic keys [17, 18].

### Molecular analyses

*In the BF area, for all the entomological surveys.* A subsample of 100 non-blood-fed *Anopheles* spp. individuals was randomly selected per survey and per village, and dissected to identify their parity state (parous or nulliparous: parous female have laid eggs at least once [19]);



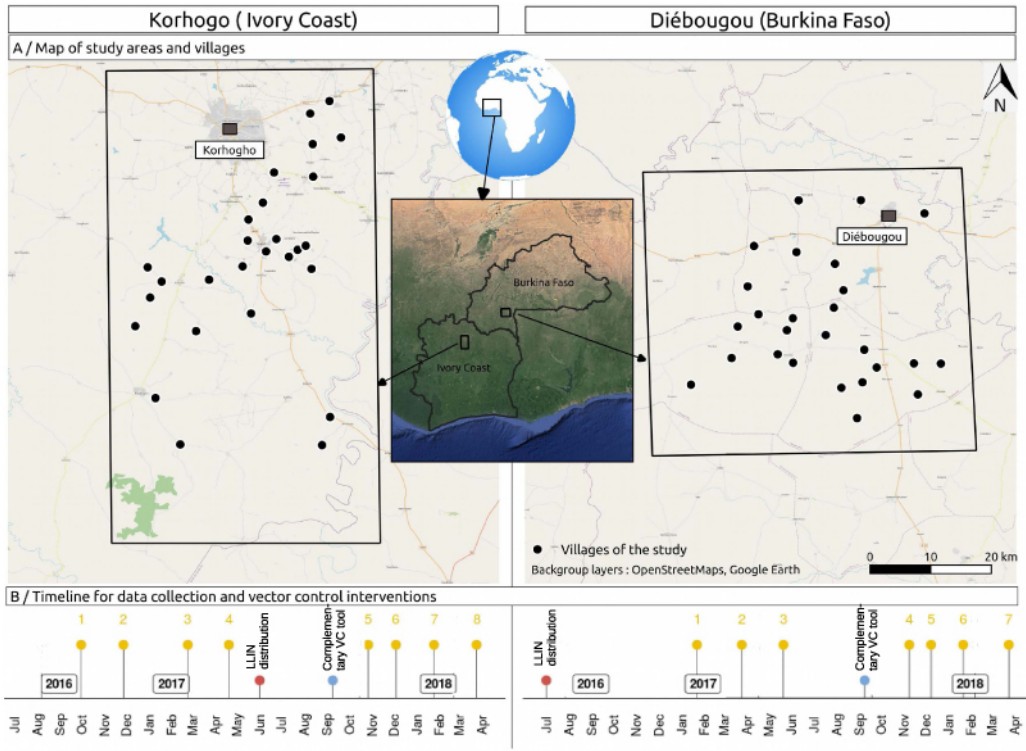

**Figure 1.** Maps of the study areas and villages, with the timeline of mosquito samplings and VC interventions. In the timeline, the entomological surveys are numbered.

- the species of all individuals belonging to the *Funestus* group or the *A. gambiae* complex were identified by Polymerase Chain Reaction (PCR);
- DNA extracted from head-thorax of individuals belonging to the *Funestus* group or the *A. gambiae* complex was used to detect *P. falciparum* infection using a quantitative polymerase chain reaction (qPCR) assay;
- PCR assays were carried out on all mosquitoes belonging to the *A. gambiae* complex to detect the L1014F (kdr-w), G119S (ace-1), and L1014S (kdr-e) mutations (kdr-w and kdr-e mutations confer insecticide resistance to pyrethroids whereas ace-1 confers resistance to carbamates and organophosphates).

*In the CI area, for the first four entomological surveys.* Due to the vast numbers of vectors collected, a subsample of *Anopheles* vectors from six villages randomly chosen out of the 28 included in the study were further analyzed:

- For all the individuals belonging to the *Anopheles nili* complex or the *A. funestus* group collected in these six villages, the *P. falciparum* infection was detected using qPCR;
- for one individual of the *A. gambiae* complex randomly selected per hour per collection site (indoors/outdoors) during each survey in these six villages: species were identified by PCR; *P. falciparum* infection was detected by qPCR; L1014F (kdr-w) and G119S (ace-1) mutations were detected by qPCR.

**Table 1.** Morphological and molecular analyses performed, with references to the methods used.

|  | Korhogo area (CI) | Diébougou area (BF) |
|---|---|---|
| PCR identification of *A. gambiae s.l.* species | [20, 21] | [22] |
| PCR identification of *A. funestus* species | Not performed | [23, 24] |
| qPCR kdr-w | [25] | [26] |
| PCR kdr-e | Not performed | [27] |
| qPCR ace-1 | [27] | [28] |
| qPCR *P. falciparum* | [29] | [29] |

**Table 2.** Key entomological metrics extracted from the data.

|  | Korhogo area (CI) | Diébougou area (BF) |
|---|---|---|
| Number of human-nights of collections | 1,854 | 1,512 |
| Number of *Anopheles* captured | 57,716 | 2,989 |
| Mean human biting rate | 31.13 bites/human/night | 1.98 bites/human/night |
| Number of *Anopheles* species captured | 6 | 10 |
| % *A. gambiae* s.s. | 97% | 20% |
| % *A. coluzzii* | <1% | 44% |
| % *A. funestus* | 1% | 24% |
| % other species | <1% | 12% |
| % outdoor bites | 56% | 42% |
| Allelic frequency of the kdr-w mutation | 89.29% | 66.83% |
| Allelic frequency of the kdr-e mutation | NA | 13.5% |
| Allelic frequency of the ace-1 mutation | 31.13% | 7.72% |
| Mean sporozoite infection rate | 2.5% | 10.9% |
| Mean entomological inoculation rate | 0.77 infected bites per human per night | 0.21 infected bites per human per night |

*In the CI area, for the last four entomological surveys.*

- For all the individuals belonging to the *A. nili* complex or the *A. funestus* group: the identification of *P. falciparum* infection was performed by qPCR;
- for a subsample representing 25% of the captured *A. gambiae*: species were identified by PCR; *P. falciparum* infection was detected by qPCR; L1014F (kdr-w) and G119S (ace-1) mutations were detected by qPCR.

References to the methods used for these molecular analyses are detailed in Table 1.

In addition to the entomological data, a set of environmental data were extracted at the places and times of sampling from Earth-observation satellite products [30–32]. The following information was extracted: % of landscape occupied by each land cover type in a 2 km radius buffer zone around the sampling site; weekly rainfall and weekly land surface temperature in a 2 km radius buffer zone around the sampling site, and up to 6 weeks before each sampling event. The methods used to generate this data are detailed in [33].

## Results

Although not the primary aim of this paper, we provide some key information extracted from the data in Table 2, including some aggregated entomological parameters of malaria transmission. Please note that detailed descriptions of the data, for the pre-intervention phase only, are available in two publications [34, 35].



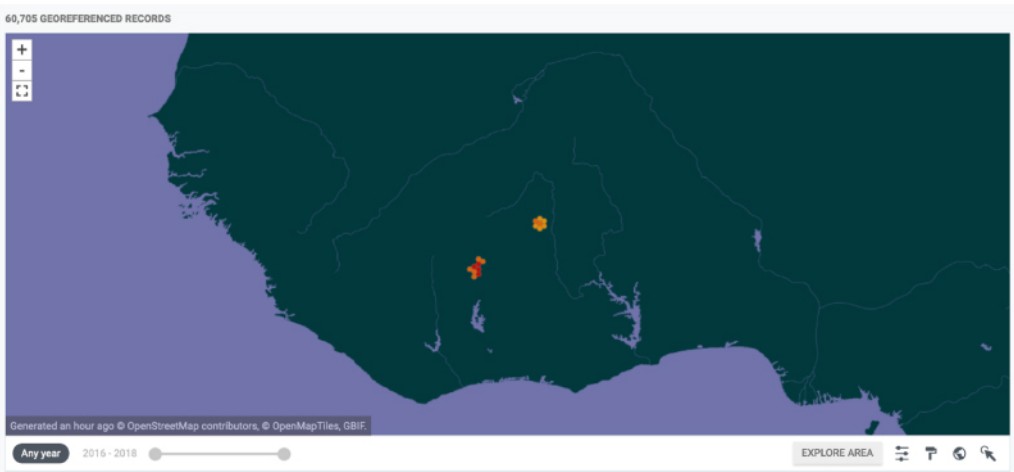

**Figure 2.** Interactive map of the georeferenced occurrences hosted by GBIF [36]. https://www.gbif.org/dataset/9f6ad2af-34cb-4f69-8436-1a41c2b43c62

## DATA VALIDATION AND QUALITY CONTROL

Each collection night, one technician, assisted by two local supervisors, supervised the mosquito collection in each village to ensure that they were correctly performed.

Independent staff supervised the rotation of the mosquito collection and regularly checked the quality of the procedure. The following criteria were checked and reported on an electronic tablet: respected collection location, collector at their post, collector awake, collector in a correct position, collector adequately dressed, and correct hourly bags used. If one of the criteria was not respected, required arrangements were immediately made by the supervisor.

All data reported here has been curated, and the terminology has been homogenized. Data has been validated using the validator available in Global Biodiversity Information Facility (GBIF) (Figure 2).

## RE-USE POTENTIAL

These data have already been used in several studies, including (non-exhaustively): a description of the entomological profiles of malaria transmission in the study areas during the pre-intervention phase of the RCT [34, 35], modelling of the environmental determinants of the biting rates of the malaria vectors in the BF area [33], modelling of the drivers of the physiological and behavioural resistance to insecticides in both study areas [37].

Examples of potential re-use include:

- comparison of entomological parameters of malaria transmission with other areas (regions, countries, continents);
- integration in global databases of vectors (e.g., the VectorBase database [38] or the Malaria Atlas Project [39]) for larger-scale studies of seasonal mosquito dynamics;
- species distribution modelling at a larger spatial scale using the environmental data.

## DATA AVAILABILITY

The data supporting this article are published through the Integrated Publishing Toolkit (IPT) of GBIF France and are available under a CC0 waiver [40]. Other data are available in the GigaDB repository [16].

The data, published as a Darwin Core archive, include four datasets:

- **Event**: the dates and geographic coordinates of the sampling events (i.e., mosquito collection);
- **Occurrence**: the collected *Anopheles* along with genus or species identification (1 row = 1 collected mosquito. We choose such a granularity to link the occurrence data with the results of the molecular analysis performed at the individual mosquito scale. Users are free to summarize the data, e.g., by EventID);
- **Extended measurement or fact**: the results of the molecular analysis for the subsample of the collected *Anopheles* species. It also includes the place of collection of each mosquito (i.e., indoors or outdoors);
- **Measurement or fact**: the environmental data (meteorological and landscape conditions) at the sampling event time-points.

The meaning of the columns of the datasets (i.e., data dictionary) is available in the List of Darwin Core terms [41].

We kindly ask users to give appropriate credit and attribution when using this data.

## EDITOR'S NOTE

This paper is part of a series of Data Release articles working with GBIF and supported by TDR, the Special Programme for Research and Training in Tropical Diseases hosted at the World Health Organization [42].

## LIST OF ABBREVIATIONS

BF: Burkina Faso; CI: Côte d'Ivoire; GBIF: Global Biodiversity Information Facility; HLC: Human Landing Catch; IPR: Institut Pierre Richet; IPT: Integrated Publishing Toolkit ; IRD: Institut de Recherche pour le Développement; IRSS: Institut de Recherche en Science de la Santé; LLIN: Long-Lasting Insecticidal Nets; PCR: Polymerase Chain Reaction; qPCR: quantitative Polymerase Chain Reaction; RCT: Randomized Controlled Trial; REACT: Insecticide resistance management in Burkina Faso and Côte d'Ivoire: research on vector control strategies; VC: Vector Control.

## DECLARATIONS

### Ethical approval and consent to participate

Ethical clearance for the study was granted by the National ethics committee in Côte d'Ivoire (No. 063/MSHP/CNER-kp) and by the Institutional Ethics Committee of the Institut de Recherche en Sciences de la Santé (No. A06/2016/CEIRES) in Burkina Faso. We received a community agreement before the beginning of the study, and we obtained written informed consent from all the mosquito collectors and supervisors. Yellow fever vaccines were administered to all the field staff. Collectors were treated free of charge when they were diagnosed with malaria during the study period, according to the recommendations of the World Health Organisation. They were also free to withdraw from the study at any time without any consequences.

## Competing Interests

The authors declare that they have no competing interests.

## Authors' contributions

PT: software, data curation, visualization, writing (original draft); BZ: supervision, investigation, data curation, conceptualization, methodology, writing (review and editing); DDS: supervision, investigation, data curation, conceptualization, methodology, writing (review and editing); LPAA: supervision, investigation, conceptualization, methodology, writing (review and editing); KM: supervision, conceptualization, methodology, funding acquisition, writing (review and editing); RKD: supervision, resources, conceptualization, methodology, funding acquisition, project administration, writing (review and editing); AAK: supervision, resources, conceptualization, methodology, funding acquisition, writing (review and editing); CP: supervision, resources, investigation, conceptualization, methodology, funding acquisition, project administration, writing (review and editing); NM: supervision, resources, investigation, data curation, conceptualization, methodology, funding acquisition, project administration, writing (review and editing).

## Funding

This work was part of the REACT project, funded by the French Initiative 5% – Expertise France (No. 15SANIN213).

## Acknowledgements

We thank all participants of the study, especially technicians at the IRSS and IPR for their technical assistance. We thank all the mosquito collectors and supervisors for their commitment to the field. We are also grateful to the villagers of all sites for their kind collaboration and hospitality. We thank Sophie Pamerlon (GBIF France) for her help with the use of IPT.

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
