## [Editor Report]

Comments to the AuthorThis Data Release presents a fine-grained dataset of potentially malaria carrying anopheles mosquitoes resulting from a series of collections in 55 villages of the rural districts of Korhogo (Northern Côte d'Ivoire) and Diébougou (South-West Burkina Faso) between 2016 and 2018. Collected using the Human Landing Catch method inside and outside households, and analyzed individually to identify species, insecticide resistance genetic mutations, plasmodium falciparum infection, and parity status. These data have already been used in several studies, but sharing it in this manner allows use in larger-scale studies such as species distribution modelling.

---

## [Reviewer Report]

Reviewer name and names of any other individual's who aided in reviewer Luis Acuña-CantilloDo you understand and agree to our policy of having open and named reviews, and having your review included with the published papers. (If no, please inform the editor that you cannot review this manuscript.)YesIs the language of sufficient quality?YesPlease add additional comments on language quality to clarify if needed
No appliedAre all data available and do they match the descriptions in the paper? YesAdditional CommentsNo appliedAre the data and metadata consistent with relevant minimum information or reporting standards? See GigaDB checklists for examples <a href="http://gigadb.org/site/guide" target="_blank">http://gigadb.org/site/guide</a>NoAdditional CommentsThe map of the study areas does not meet design requirements, it must be broader from the continent, the country to the study areas. The scales and compass are missing. The timeline is very small, when you enlarge the image it loses resolution. In table 2, the names of the species must be written in italics
Is the data acquisition clear, complete and methodologically sound?YesAdditional CommentsIs there sufficient detail in the methods and data-processing steps to allow reproduction?YesAdditional CommentsYes, but it is important to describe the PCR methodology used in the Korhogo area to identify mosquitoes of the An.funestus species and the kdr-e geneIs there sufficient data validation and statistical analyses of data quality? Not my area of expertiseAdditional CommentsIs the validation suitable for this type of data?YesAdditional CommentsIs there sufficient information for others to reuse this dataset or integrate it with other data?YesAdditional CommentsAny Additional Overall Comments to the AuthorThey must unify the writing of scientific names, all in italicsRecommendationMinor Revision

---

## [Reviewer Report]

Reviewer name and names of any other individual's who aided in reviewer Angeliki MartinouDo you understand and agree to our policy of having open and named reviews, and having your review included with the published papers. (If no, please inform the editor that you cannot review this manuscript.)YesIs the language of sufficient quality?YesPlease add additional comments on language quality to clarify if needed
Are all data available and do they match the descriptions in the paper? YesAdditional CommentsAre the data and metadata consistent with relevant minimum information or reporting standards? See GigaDB checklists for examples <a href="http://gigadb.org/site/guide" target="_blank">http://gigadb.org/site/guide</a>YesAdditional CommentsIs the data acquisition clear, complete and methodologically sound?YesAdditional CommentsIs there sufficient detail in the methods and data-processing steps to allow reproduction?YesAdditional CommentsIs there sufficient data validation and statistical analyses of data quality? YesAdditional CommentsIs the validation suitable for this type of data?YesAdditional CommentsIs there sufficient information for others to reuse this dataset or integrate it with other data?YesAdditional CommentsAny Additional Overall Comments to the AuthorThe paper is really well written and the amount of work is impressive.  I would add in the abstract that the Human Landing Catch was performed by experts. Since Human landing catches should only be done by experts. The figure with the map showing the villages needs to be re-drawn as the moment the graphics are of poor quality. I can not actually find the file with the data itself RecommendationAccept

---

## [Reviewer Report]

Upload additional filesDRR-202305-03/form/DRR-202305-03_Data-Review-MAT.pdfReviewer name and names of any other individual's who aided in reviewer Mary Ann TuliDo you understand and agree to our policy of having open and named reviews, and having your review included with the published papers. (If no, please inform the editor that you cannot review this manuscript.)YesIs the language of sufficient quality?YesPlease add additional comments on language quality to clarify if needed
The paper is very well written. It is comprehensive, well organised and easy to read.Are all data available and do they match the descriptions in the paper? YesAdditional CommentsThe dataset had been made available in GBIF: https://doi.org/10.15468/v8fvynAre the data and metadata consistent with relevant minimum information or reporting standards? See GigaDB checklists for examples <a href="http://gigadb.org/site/guide" target="_blank">http://gigadb.org/site/guide</a>YesAdditional CommentsIs the data acquisition clear, complete and methodologically sound?YesAdditional CommentsIs there sufficient detail in the methods and data-processing steps to allow reproduction?YesAdditional CommentsIs there sufficient data validation and statistical analyses of data quality? YesAdditional CommentsIs the validation suitable for this type of data?YesAdditional CommentsIs there sufficient information for others to reuse this dataset or integrate it with other data?YesAdditional CommentsAny Additional Overall Comments to the AuthorRecommendationAccept

---

## [Reviewer Report]

Reviewer name and names of any other individual's who aided in reviewer Gerard RyanDo you understand and agree to our policy of having open and named reviews, and having your review included with the published papers. (If no, please inform the editor that you cannot review this manuscript.)YesIs the language of sufficient quality?NoPlease add additional comments on language quality to clarify if needed
The formatting of species names is inconsistent and often not matching accepted standards for presentation of scientific names. Genus and species should always be italicised, and the genus capitalised (e.g. see https://en.wikipedia.org/wiki/Binomial_nomenclature#Writing_binomial_names) I have identified locations below that need attention. line 43, 94 - Anopheles should be italicised. 44 - Catches not Catch 46 - plasmodium falciparum should be italicised and with capital P 49, 81, 239, table 2 - anopheles should be italicised and capital A Table 2 - species should all be italicisedAre all data available and do they match the descriptions in the paper? YesAdditional CommentsAre the data and metadata consistent with relevant minimum information or reporting standards? See GigaDB checklists for examples <a href="http://gigadb.org/site/guide" target="_blank">http://gigadb.org/site/guide</a>NoAdditional CommentsI'm not certain here. I could not see metadata describing the content and unit of each column of data housed with the data, though that may be due to my unfamiliarity with interacting directly with the gbif website, and I can't really make out what's in some of the downloaded xmls. This paper does not contain a metadata description in the tabular format, though most of the "required metadata" per link is contained textually throughout the paper. Is the data acquisition clear, complete and methodologically sound?NoAdditional CommentsThe intervention type is not described - what were the complimentary VC strategies and when and how were they administered across space? Which were the control and experimental villages?  line 163 - was this done for all Anopheles, or a sub sample?
Is there sufficient detail in the methods and data-processing steps to allow reproduction?NoAdditional CommentsIs there sufficient data validation and statistical analyses of data quality? YesAdditional CommentsNAIs the validation suitable for this type of data?YesAdditional CommentsNAIs there sufficient information for others to reuse this dataset or integrate it with other data?YesAdditional CommentsAny Additional Overall Comments to the AuthorOverall a nice and valuable contribution.  Figure 1 is not labelled as figure 1, and this quality of the image is poor and difficult to read.RecommendationMinor Revision